# Prevalence of *Toxocara* Eggs in Public Parks in the City of Valencia (Eastern Spain)

**DOI:** 10.3390/vetsci9050232

**Published:** 2022-05-11

**Authors:** Belinda Rose Köchle, María Magdalena Garijo-Toledo, Lola Llobat, José Sansano-Maestre

**Affiliations:** 1Department of Animal Production and Public Health, Faculty of Veterinary Medicine and Experimental Sciences, Universidad Católica de Valencia San Vicente Mártir, Calle Guillem de Castro, 94, 46003 Valencia, Spain; riucv@ucv.es; 2Department of Animal Production and Health, Public Veterinary Health and Food Science and Technology, Faculty of Veterinary Medicine, Universidad Cardenal Herrera-CEU, CEU Universities, Calle Tirant Lo Blanc, 7, 46115 Valencia, Spain; maria.llobatbordes@uchceu.es

**Keywords:** parasitic zoonoses, public parks, pets, Spain, *Toxocara*

## Abstract

*Toxocara* spp. is one of the most common zoonotic geohelminths in the world. Its infections are associated with the accidental ingestion of contaminated soil and affecting, especially children. In this study, feces, and soil samples from 14 public parks in the city of Valencia were analyzed. The Telemann method and a modified version of a sieving technique were used to process feces and soil, respectively. None of the fecal samples and 10.9% of soil samples from five parks (35.7%) tested positive for the presence of *Toxocara* eggs. The most contaminated areas were the canine sanitary parks (30.8% of the samples), followed by socialization areas for dogs (9.7%); no positive samples were found at children’s playgrounds. Our results suggest that most pets in Valencia are periodically dewormed, although additional preventive measures should be applied, since the risk of infection exists probably due to the presence of stray dogs and feral cats.

## 1. Introduction

The most prevalent zoonoses related to geohelminths around the world are *Toxocara* spp., *Ancylostoma* spp., and *Strongyloides* spp., which are responsible for more than two billion infections in people around the world [1], as well as important costs associated with these infections [2]. Nematodes of the genus *Toxocara* are responsible for human toxocariasis (HT), one of the five neglected parasitic diseases with priority for public health action [3]. This genus includes two species, *T. canis* and *T. cati*, whose definitive hosts are dogs and cats, respectively. Infected animals shed eggs with their feces in the environment, where they can remain viable for years in shady, humid soils and at cool temperatures [4,5]. Humans, especially children, are accidental hosts when exposed to contaminated soils via the fecal-oral route [6,7]. Several clinical forms of HT are described: (1) covert/common (the most frequent), (2) visceral larva migrans (VLM), and (3) ocular larva migrans (OLM) [8]. Non-specific symptoms, such as fever, abdominal pain, and asthma, can be seen with covert toxocariasis [9,10]. VLM and OLM are both primarily diagnosed in young children, the first one including clinical manifestations related to inflammation of the internal organs, such as asthma or myelitis [11,12,13,14,15,16], and cutaneous reactions [17]. OLM runs with visual loss, strabismus, or retinal granuloma [8,12]. Occasionally, the central nervous system can be affected in middle-aged patients, which is called neurotoxocariasis [18,19], and has been linked to meningoencephalitis, epileptic seizures, and neurodegenerative disorders [20,21,22,23,24,25,26].

Public parks and sandpits are an important source of infection for children, since dogs and cats share public areas with them [27]. A fifth of the public places in the world are contaminated with *Toxocara* eggs, with prevalence rates ranging from 13 to 35%, depending on the geographical area [5]. In the Iberian Peninsula, the prevalence in public parks varies between 16.4% in Madrid [28], 37% in Tenerife [29], 45.5% in Córdoba [30], 50% in Lisbon [31], and 67% in Murcia [32]. The main objective of this study is to determine the presence of *Toxocara* eggs in soil samples and dog feces recovered in public areas with different human and animal presence in the city of Valencia, in order to suggest the risk of human infection with these zoonotic species, and to establish more effective prevention and control measures than those that are used currently.

## 2. Materials and Methods

### 2.1. Study Design and Collection of Samples

The study was carried out in 14 public parks located within the most populated zones of the city of Valencia (in the east of the Iberian Peninsula), with an estimated population of about 800,000 inhabitants (Figure 1). The selection criteria were the presence of canine socialization areas (SO), canine sanitary areas (SA), and/or children’s playgrounds (CP) (Figure 2). The study was carried out between November 2018 and June 2019. Fecal and soil samples were randomly collected from the three types of areas. Initially, the most recent feces were taken randomly from the ground. After that, the surface area was divided into transects according to the extension, and soil samples were taken systematically along transect lines approximately 10 m apart. Soil samples were collected with a garden shovel, at a depth of 2 to 5 cm on a 10 cm surface and weighing approximately 200 g each. Only stool samples with a fresh appearance were selected. The number of samples varied according to the park size. Sampling in the SO with a large extension started in a corner of the enclosure, and progressed through transects until the entire study surface was covered, picking up samples with a minimum distance of five meters between the points of collection. In the CP with rubber flooring, soil samples were taken from the perimeter. To minimize time-conditioned selection bias, sampling was carried out during three different moments of the day: morning, midday, and afternoon. All samples were stored in individual bags at 4 °C until they were analyzed, for a maximum of two days. Each bag was marked with the following characteristics: date of collection, name of the park, type of sample (sand, feces), and park area (SO, SA, CP).

### 2.2. Feces Analysis

Fecal samples were first macroscopically examined to detect color, consistency, and the presence or absence of mucus, blood, or fibrin. Then, an adaption of the Telemann method was used to recover helminth eggs [33]. Next, 3 g of feces were weighed and mixed with 5% acetic acid at a ratio of 1/5 using a mortar. The mixture was filtered through a strainer with double gauze, and 5 mL were mixed with 5 mL of diethyl ether and centrifuged for 5 min at 500× *g*, after which the supernatant was discarded. Next, 10 ml of zinc sulphate solution (specific gravity 1.2 g/cm^3^) was added to each tube, vortexed, and centrifuged for 5 min at 500× *g*. Finally, tubes were filled with the same solution to form a positive meniscus, and a coverslip was placed on them. After 15 min, samples were examined with an optical microscope.

### 2.3. Soil Analysis

Soil samples were analyzed following a modified version of a sieving technique [32,34]. A measure of 100 g of sample was washed with running tap water, and each sample was sifted through four metal sieves with decreasing pore diameters (1000 µm, 250 µm, 125 µm, and 63 µm) to gradually eliminate the bigger particles. The residue in the last sieve was flushed into a 1000 mL sedimentation cup filled with water. After 20 min of sedimentation, the supernatant liquid was discarded, and the wash was repeated two more times. After the last wash, the sediment was collected into 15 mL tubes for centrifugation for 5 min at 500× *g*. The supernatant was discarded, and then 10 mL of zinc sulphate solution (specific gravity 1.2 g/cm^3^) were added to each tube, vortexed, and centrifuged for 5 min at 500× *g*. Finally, tubes were filled with the same solution to form a positive meniscus and a coverslip was placed on it. After 15 min, samples were examined with an optical microscope. In both coprological and soil analysis, samples were classified as positive or negative, but only samples with viable eggs were considered positive [4]. Due to the close similarity between eggs of *Toxocara canis* and *T. cati*, no attempt was made to differentiate between them.

### 2.4. Statistical Analysis

Statistical analysis was performed with statistical package R Commander and RcmdrPlugin. The 95% confidence intervals for prevalence estimates were calculated using the Wilson score interval method. The association between *Toxocara* spp. presence and categorical factors (park, zone, and type of sample) was compared using Pearson’s χ^2^ test, and the confidence intervals for prevalence estimates were calculated using the Wilson score interval method. A *p*-value < 0.05 was reported as statistically significant.

## 3. Results

The results of the research are presented in the summary Table 1. A total of 108 samples (64 soil and 44 fecal) from 14 public parks were analyzed. Of them, seven soil samples (10.9%) and none of the fecal samples were contaminated, showing a positive relationship between the type of sample and the presence of *Toxocara* eggs (*p* < 0.05) (Figure 3). Stools were identified as dog origin, according to size, aspect, and location, as they were not covered by a substrate. After macroscopical examination, all the fecal samples showed a normal color and consistency, with no presence of mucus, blood, or fibrin. Positive samples were collected from 5/14 (35.7%) different parks. Sanitary areas showed the highest contamination rate, with 30.8% (4/13) of samples being positive, followed by SO for dogs (9.7%; 3/31). All samples collected at CP were negative. No significant differences were found between the examined parks or between the three types of areas.

## 4. Discussion

In the last decade, both the number of pet cats and pet dogs has increased significantly worldwide [35]. In urban areas, public parks with canine sanitary areas, socialization areas for dogs, and children’s playgrounds are very common. These areas, in which animals defecate, are shared with humans, and represent a serious risk of zoonoses, especially for those children with pica habits. *Toxocara* spp. is one of the most prevalent zoonotic geohelminths that can cause severe pathologies in humans. In this survey, the contamination with *Toxocara* spp. eggs of soil and fecal samples collected from 14 public parks in the city of Valencia was studied.

Surprisingly, none of the examined fecal samples were positive for the presence of *Toxocara* eggs, most likely due to the majority being owned as pets. In Spain, the helminth zoonoses prevention programs in the last years have resulted in better dog pet management, including periodical deworming and feed improvement [36]. Kutdang et al. (2010) realized that mixed dog breeds, more frequent in the stray group, had higher infection rates than exotic breeds [37]. Stray animals are also more susceptible to being infected by ingesting paratenic hosts, such as rodents, which are often carriers for infective larvae [38,39]. Furthermore, dog owners have changed their awareness about cleaning their animals’ feces, which contributes to reducing the environmental contamination. Accordingly, although there are not many studies similar to the present one in Spain, and considering that it is not possible to compare this study with others due to the different sampling and detection methods, only 1.3% of fecal samples from Madrid had ascarid (*Toxascaris leonina*) eggs [28]. Other studies, such as the one carried out in Murcia, showed that the risk of infection with intestinal helminths was significantly higher in stray than in household dogs [33]. In addition, Martínez-Moreno et al. (2007) found *T. canis* eggs in 17.7% of stray dogs in Córdoba [30].

In other European countries, the infection of feces with *T. canis* oscillated between 3% in stray dogs from Serbia [40], to 5% in the Greater Lisbon area (Portugal) [31], and 23.4% in Poland [41]. Data from some other surveys related the infection with *T. canis* to the presence of ownerless dogs in South Africa [42], Nigeria [43], and Mexico [44].

Regarding soil samples, we found an infection rate of 10.9%, higher than in other regions in Spain. For instance, in Córdoba, the prevalence was 3.8% [30], and in Murcia, it was 1.24% [32]. These results are in line with the metanalysis carried out by Fakhri et al. (2018), in which the mean prevalence of *Toxocara* spp. in soil samples in Spain was estimated at 5–8% [5]. In Madrid, however, 16.4% of soil samples were contaminated with *Toxocara* spp. [28]. In general, data from previous studies in Spain were lower than 18% of the pooled prevalence in Europe [5]. As is already known, ascarid eggs die with temperatures higher than 37 ºC, low relative humidity, and direct exposure to the sun [45]. Conversely, higher prevalence has been significantly associated with high geographic longitude, low latitude, low temperature, and high relative humidity [4,5]. Marked fluctuations in relative humidity were registered throughout the months of the present study, with the sampling period being the period with the lowest relative humidity of the year (mean of 60%, AEMET, 2018). According to this, the average temperatures in Valencia are lower than in Murcia and Córdoba, but higher than those registered in Madrid and other European countries. It is also important to consider that sampling was carried out during autumn, winter, and spring, but not during the hot and dry summer months when egg mortality increases in the environment. Thus, it would be expected to find an even lower prevalence in this study. In any case, sand soil represents a threat of human infection, not only for children playing in parks, but because eggs in the soil can reach the homes transferred from animal’s feet, as well as from the soles of people’s shoes [46]. Furthermore, it has been demonstrated that a single egg is enough to cause HT in an immunocompromised human being [47,48].

Considering the number of the analyzed parks, five (35.7%) were contaminated, a value within 16.4% of positive areas for the presence of *Toxocara* spp. in Madrid [29] and 45.5% for ascarids (*Toxocara* and *Toxascaris*) in Córdoba [30]. A recent study in New York showed that 100% of the public spaces were infected with *Toxocara* spp., with most of the eggs being identified as *T. cati* [49]. The number of feral cats in public parks in Spain is quite low compared with other countries such as the United States and, although we did not identify the species, a high density of cats in public parks may contribute to a high prevalence of *Toxocara* spp. eggs [5]. In this sense, cats can easily access every location of the park, even fenced areas, by jumping. In addition, dog feces are usually removed by the owners, while cats usually bury theirs, contributing to soil contamination [50]. Furthermore, while cats can shed *T. cati* eggs throughout their lives, dogs are more commonly infected when they are puppies [51,52,53,54]. We did not attempt to differentiate species in our study, since both are zoonotic and equally important in the study, and discrimination by optical microscope is not easy [55].

Sanitary areas for dogs were the most contaminated. These results were expected, since they are the most frequented ones by animals. Although the study’s findings suggest that pets in these areas of Valencia are well dewormed, owners are not used to removing feces, so eggs can reach the soil easily. Furthermore, the presence of stools attracts other stray animals (including dogs and cats) to defecate in the same place [54,56]. Socialization areas presented a lower prevalence for *Toxocara* spp. eggs. A high number of pets can be found in these places throughout the day. However, contrary to what has been described for sanitary areas, pet owners are more likely to remove fecal droppings for their animals to play in more hygienic conditions. Signs are present at the entrance of these areas specifying the obligation of owners to collect their pet’s feces into bags, and it is also common to find brooms, dustpans, and bag dispensers for the removal and disposal of animal feces. On the other hand, no positive samples were found at the children’s playgrounds, probably due to the preventive measures adopted by the council, including the prohibition of the access animals to them. Furthermore, most of these areas are fenced, avoiding the entry of animals. Moreover, floors are comprised of rubber material, and so the possibility of animal infection by contaminated soil is reduced.

The fact that we did not find eggs in the feces, but did in the soil, is consistent with the results of previous studies in which prevalence rates for soil samples were higher than for fecal samples [28,31]. As mentioned above, the positive soil samples probably were from stray dogs, but also may have been from feral cats and cat feces, representing a more important potential source of environmental contamination with zoonotic parasites than dog feces [57]. On the other hand, fecal analysis in the study represented animals sampled at one moment, while soil analysis may represent the accumulated concentration of more than one animal’s feces over a period, considering the high resistance of *Toxocara* spp. eggs in the environment [58].

## 5. Conclusions

Ascarids are common parasites in dogs and cats, and in this study, we found them in soil samples taken from several public parks examined in the city of Valencia, representing a risk of zoonosis. As is already known, the infection may become effective not only by hands-on contact with sand, but also by transferring the eggs to peoples’ houses on feet and shoes. Thus, our results suggest a more exhaustive control on the contamination of soil-transmitted helminths in public areas, as well as improved preventive measures. Some of the measures recommended to decrease the threat of toxocariasis in children—for whom it is especially dangerous—may include the following: elimination or reduction of intestinal infections in definitive hosts by regular anthelmintic treatments and fecal testing, fencing of sanitary and socialization areas to avoid the access of stray dogs, control of stray animals, removing feces from the soil, substitution of sand soil by a rubber or other compact material in walking areas, and educational programs for the public and one health approach.

## Figures and Tables

**Figure 1 vetsci-09-00232-f001:**
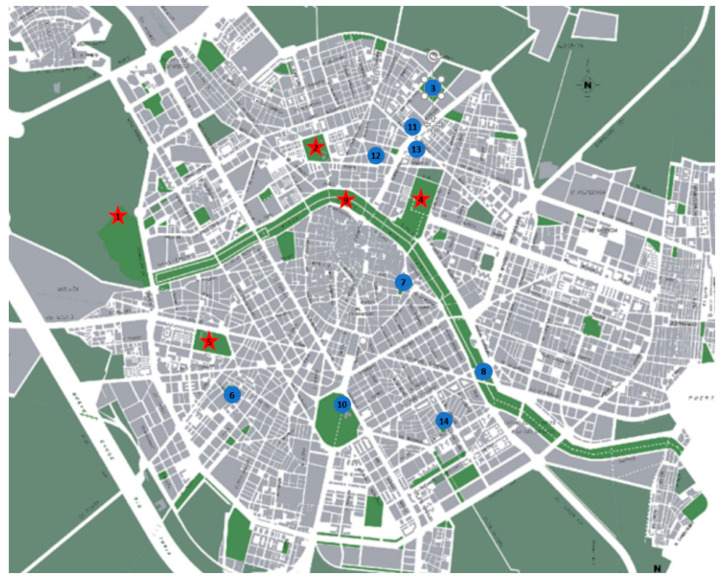
Map of the metropolitan area of Valencia, highlighting the positive (red stars) and negative (blue circles) parks to *Toxocara* spp. eggs. Numbers inside correspond to references to the park according to Table 1.

**Figure 2 vetsci-09-00232-f002:**
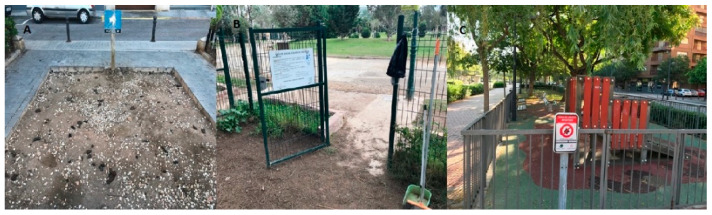
Types of studied areas. (**A**): canine sanitary area, (**B**): socialization area, (**C**): children’s playground.

**Figure 3 vetsci-09-00232-f003:**
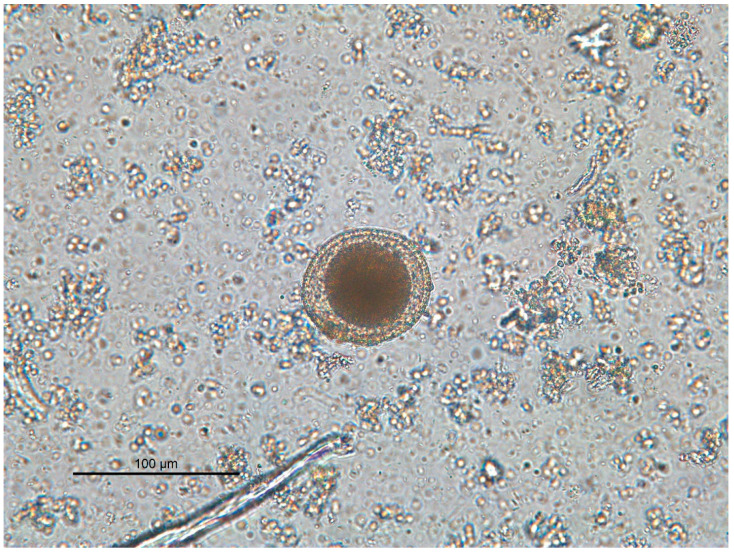
*Toxocara* spp. viable non-infective egg.

**Table 1 vetsci-09-00232-t001:** Prevalence of *Toxocara* spp. eggs in soil and feces from parks in Valencia. Distribution of positive samples regarding the area of collection.

	Reference Number	Park Name	+ *Toxocara* spp. Egg	Sampling Area *	N° Soil Samples	+ Soil Samples (%)	N° Feces Samples	+ Feces Samples
Public park (PP)	1	Parque de Cabecera	+	SA	0	0	1	0
				SO	8	2 (25)	6	0
	2	Parque de Marxalenes	+	CP	2	1 (50)	1	0
	3	Parque de Orriols	−	CP	1	0	1	0
	4	Jardines del Real (Viveros)	+	SA	0	0	3	0
				CP	4	2 (50)	1	0
	5	Parque del Oeste	+	SA	6	1 (16.7)	2	0
				CP	2	0	2	0
	6	Pl. Enrique Granados	−	SA	6	0	4	0
	7	Jardines de la Glorieta	−	SA	1	0	3	0
				CP	1	0	0	0
	8	Tramo XII del Cauce del río Turia	−	SO	7	0	1	0
				CP	1	0	1	0
	9	Tramo VI del Cauce del río Turia	+	SO	10	1 (10)	4	0
	10	Parque Central	−	SO	5	0	2	0
	11	Ludoparc Salka	−	CP	5	0	3	0
	12	Parque Galp	−	CP	2	0	2	0
	13	Pl. de Manuel Laguarda Cubell	−	CP	2	0	2	0
	14	Pl. Dr Torrens	−	SO	3	0	3	0
Total PP		14	35.7% (5/14)		64	10.9% (7/64)	44	0
Total SO		38	+		22	3	16	0
Total SA		47	+		29	3	18	0
Total CP		23	+		13	1	10	0

+: samples with *Toxocara* eggs; −: samples without *Toxocara* eggs. * SO: socialization area; SA: sanitary area; CP: children’s playground.

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
