# Peer review of "Prevalence of Toxocara Eggs in Public Parks in the City of Valencia (Eastern Spain)"

_vetsci, 2022, doi:10.3390/vetsci9050232_

Round 1
Reviewer 1 Report
This is an interesting article which adds to the data on soil contamination with Toxocara eggs from canid or felid origin. It seems well laid out and well executed and is generally well written. The data is in line with other studies.
I have a few minor comments below, most of which are just grammatical suggestions. I have tried to offer the suggestions of the way to reword it
Line 13- maybe faecally contaminated rather than accidentally contaminated?
Line 18- delete ‘of’
Line 30- faeces in the environment rather than on?
Line 32- children rather than child
Line 32- accidental hosts when they are infected by coming into contact with contaminated soil ….. (reword)
Line 35- abdominal pain and asthma (reword)
Line 62- where you talk about stool samples, where were they collected from? The same places? The ground? Dog waste bins? Close to where the soil was collected?
Line 80- you don’t really talk much about colour or consistency in the document. But how were these graded?
Line 83- The mixture was filtered through …. (reword)
Line 142- also more susceptible to being infected by …. (reword)
Line 143-144- I cannot work out what you are trying to say here so it is difficult for me to suggest a rephrasing, but please rephrase this line
Line 159- delete ‘a’
Line 183- delete ‘to’
Line 190- which makes senses since they are the most … (Reword)
Line 192- delete ‘to’
Line 192-193- This doesn’t make sense and I am not sure what to suggest to rephrase. Please modify it
Line 194- can be found throughout the day …. (Reword)
Line 197- collect is typed incorrectly
Line 204- remarkable that none of the stool samples …. (reword)
Author Response
Dear reviewer,
Enclosed you can find the responses to your notes.
Thank you very much in advance.
Kind regards

Reviewer 2 Report
The article, Prevalence of Toxocara eggs in public parks of the city of Valencia (Eastern Spain)" presents and discusses the findings after investigating Toxocara spp. presence in public playgrounds, dog socialization and sanitation areas in parks and suggests preventative actions to be adopted for authorities in order to reduce the risk of zoonotic infection .Overall, the article is of local /regional importance and could be considered for publication but only after changes are made.
Minor points
The entire article needs to be reviewed by a scientifically orientated English speaking native as many grammatical errors are found throughout the text.
Line 13: " accidental ingestion of contaminated soil"
Lines 13-14: "In this study, soil and faecal samples…"
Line 14:"10.9% of samples….."
Line 18:" most pets in Valencia.."
Line 32-33:" Humans, especially children, are accidental hosts when infected via the faecal-oral route when exposed to contaminated soils [6,7]. "
Lines 33-35: "Several clinical forms of HT are described: 1) covert/common (the most frequent),2) Visceral Larva Migrans (VLM) and 3) Ocular 34 Larva Migrans (OLM) [8]."
Lines 39-40: ". Occasionally, the central nervous system can be affected in middle-aged patients"
Lines 82-83:" Three grams of faeces were weighed and mixed with 5% acetic acid at a ratio of 1/5 using a mortar. The mixture was filtered… "
Line 99:" vortexed and centrifuged"
Lines 100-101: "placed on them".
Lines 116-117:" Positive samples were collected from 5/14 ( 35.7% ) different parks."
Line 118:" Sanitary areas showed the highest contamination rate"
Line 121 :" types of areas".
Table 1 is too squashed as is and will need to be placed entirely onto one page to be understood. If this is impossible, at least the second page should have legend categories placed again as on the first page as to not make the table confusing for the reader. Maybe the second part of the same table starting with totals can be placed as new table.
Lines 128-129:" significantly worldwide [35]"
Line 131:" represent a serious risk of zoonoses"
Line 137:" results that were expected since most of them were probably from owned pets"
Lines 138-139:" in the last years have resulted in better dog management, including periodical deworming and feed improvement [36]".
Line 143-144: Change sentence "Besides, pet ….. contamination" to make sense.
Lines 155-156: The sentence is not clear. Is it other counties in Spain or other European countries? If the latter, then please write it as " Regarding soil samples, we found an infection rate of 10.9% in Spain, higher than in other countries".
Lines 179-180:" most of the eggs identified as T. cati [49]."
Line 187:" when they are puppies"
Lines 188-189:" both are zoonotic and equally important in the study, and discrimination by optical microscope is not easy [55]"
Change the sentence in lines 190-193 to be grammatically correct. Also, there appears to be a contradiction by relating these areas to be more contaminated with this statement as earlier in the manuscript (lines 136-137) it was hinted that most dogs were treated with anthelmintics and so even dogs in sanitary places are of the same dewormed denominator and should not increase in the pool of helminth eggs. Maybe, cats that shed T. cati play a larger role in this contamination.
Line 192: Perhaps, the following was meant? " Besides, the presence of stools attracts other …"
Line 194:" high number of pets can be found during the day"
Lines 197-198:" Signs are present at the entrance to these areas specifying the obligation of owners to collet their pet faeces into bags and it is also common to find brooms, dustpans, and bag dispensers for the removal and disposal of animal feces".
Line 201:" avoiding the entry of animals"
Line 202:" Also, floors are made of rubber and so the possibilities for infection by contaminated soil"
Lines 209-212:" Faecal analysis in the study represented animals sampled at one moment, while soil analysis may represent the accumulated concentration of more than one animal’s faeces along a period of time, considering the high resistance of Toxocara spp. eggs in the environment [58]."
Lines 214-225: The conclusion paragraph has all important points as a take home message but should be rewritten after being reviewed by a native English speaking individual in order to make sense.
Author Response

(The authors gave the same response as above.)

Reviewer 3 Report
First of all, the subject is very important and will arouse interest in both the veterinary and human medical community.
But, the microscopy method used in the study is quite old and it is known to have low sensitivity to fecal parasites. For this reason, PCR or rapid test methods must be added to this study to increase sensitivity.
In addition, fecal sampling methods are very confusing in this study design. Because fecal samples were taken from the ground, It can be confused whether the stool samples are from dogs or cats. Also, I wonder whether the dogs are stray dogs or owned dogs.
My other corrections are added to file

Author Response

(The authors gave the same response as above.)

Round 2
Reviewer 2 Report
I suggest accepting the article after more revision by an English speaker and addressing the following minor points:
Minor points:
Abstract
Lines 14-15: "Toxocara spp. is one of the most common zoonotic geohelminths in the world, its infections associated with accidental ingestion of contaminated soil and especially affecting children."
Line 17: " technique"
Line 18: Space between "and 10.9%.."
Line 28: "and important costs associated with them".
Line 31: Place comma after "two species".
Line 45: " A fifth of the public places in the world are…."
Line 47: " Place comma after Peninsula.
Lines 51-53: ".. in the city of Valencia, in order to suggest risk of human infection with these zoonotic species and to establish more effective prevention and control measures than those that are used currently."
Line 61: " Initially, the most recent faeces…."
Line 82: Remove comma between "presence or absence of mucus"
Line 108: Place comma after T. cati"
Table 1 is too cramped and it is suggested to orient it on a separate page vertically or alternatively, the legends "Positive fecal samples" and 'Positive sand samples" can be shortened to "Pos. fecal" or "Pos. sand samples" or "+ sand samples" or "+ fecal samples" with definition in the legend that "+ or Pos. is Positive" in order to properly write the legends in the table.
Lines 142-143: "Surprisingly, none of the examined faecal samples were positive for the presence of Toxocara eggs, most likely due to the majority being owned pets."
Lines 152-154: " … with others because of the different sampling and detection methods, only 1.3% of fecal samples from Madrid had ascarid (Toxascaris leonina) eggs [28]."
Line 159: "Greater"
Line 167: "In general, data from previous studies in Spain was lower than 18% of the average in Europe [5]." Try explain exactly what "the average" is. Is it the cumulative average?
Line 169: " Direct exposure to the sun"
Line 170: What is " higher geographic longitudes?"
Lines 170-173: ". Marked fluctuations in relative humidity were registered throughout the months of the present study, with the sampling period being the period with the lowest relative humidity of the year"
Line 192: " while cats usually bury them"
Lines 198-199: " Although the study's findings suggest that pets in these areas of Valencia are well dewormed, owners are not used to removing faeces…"
Lines 202-205: "A high number of pets can be found in these places throughout the day. However, contrary to what has been described for sanitary areas, pet owners are more likely to remove faecal droppings in order for their animals to play in more hygienic conditions".
Lines 209-210: ", including prohibition of access of these animals to them".
Line 211: " Also, floors are comprised of rubber material and so avoid the possibility of animal infection by contaminated soil."
Lines 212-213: I don’t understand. Is it " Remarkably, none of the stool samples examined was positive in this study, which is similar to data from previous literature?"
Line 214: "samples probably were…."
Change 224-227 to make sense.
Author Response
Dear reviewer,
Thank you very much for your comments and corrections. Enclosed you can find the new version, with all the changes made.
Kind regards,

Reviewer 3 Report
The authors made the necessary corrections. It is appropriate to publish.
Author Response
Dear reviewer,
Thank you very much for your kindly revision of the manuscript.
Kind regards,